

# Negative and positive interspecific interactions involving jellyfish polyps in marine sessile communities

Jade Boughton[1], Andrew G. Hirst[2,3], Cathy H. Lucas[4] and Matthew Spencer[5]

[1] Faculty of Sciences, International Master of Science in Marine Biological Resources (Consortium, EMBRC), University of Ghent, Ghent, Belgium
[2] School of Animal, Rural and Environmental Sciences, Brackenhurst Campus, Nottingham Trent University, Southwell, United Kingdom
[3] Centre for Ocean Life, National Institute for Aquatic Resources, Technical University of Denmark, Charlottenlund, Denmark
[4] National Oceanography Centre, University of Southampton, Southampton, United Kingdom
[5] School of Environmental Sciences, University of Liverpool, Liverpool, United Kingdom

Corresponding author
Matthew Spencer,
m.spencer@liverpool.ac.uk

## ABSTRACT

Sessile marine invertebrates on hard substrates are one of the two canonical examples of communities structured by competition, but some aspects of their dynamics remain poorly understood. Jellyfish polyps are an important but under-studied component of these communities. We determined how jellyfish polyps interact with their potential competitors in sessile marine hard-substrate communities, using a combination of experiments and modelling. We carried out an experimental study of the interaction between polyps of the moon jellyfish *Aurelia aurita* and potential competitors on settlement panels, in which we determined the effects of reduction in relative abundance of either *A. aurita* or potential competitors at two depths. We predicted that removal of potential competitors would result in a relative increase in *A. aurita* that would not depend on depth, and that removal of *A. aurita* would result in a relative increase in potential competitors that would be stronger at shallower depths, where oxygen is less likely to be limiting. Removal of potential competitors resulted in a relative increase in *A. aurita* at both depths, as predicted. Unexpectedly, removal of *A. aurita* resulted in a relative decrease in potential competitors at both depths. We investigated a range of models of competition for space, of which the most successful involved enhanced overgrowth of *A. aurita* by potential competitors, but none of these models was completely able to reproduce the observed pattern. Our results suggest that interspecific interactions in this canonical example of a competitive system are more complex than is generally believed.

## INTRODUCTION

The two canonical examples of communities structured by competition are sessile marine invertebrates on hard substrates (usually thought to be structured by competition for space) and terrestrial vertebrates (usually thought to be structured by exploitation

competition for food) (*Roughgarden, 1986*). These examples are distinct because opportunities for niche partitioning of space are limited, while resources such as food can generally be partitioned in ways that enhance coexistence (*Yodzis, 1978*, pp. 8–10). Another key difference between these two canonical examples is that marine sessile communities are often modelled as open systems, while terrestrial vertebrate communities are often treated as closed (*Roughgarden, 1986*). In consequence, marine sessile communities such as rocky shores, coral reefs and subtidal encrusting and fouling communities have played a key role in the development of theory including the importance of competition in determining distributions (*Connell, 1961*), the existence of alternative stable states (*Sutherland, 1974*), non-transitive networks of interactions (*Buss & Jackson, 1979*), mathematical models of open systems (*Roughgarden, Iwasa & Baxter, 1985*) and Markov models (*Hill, Witman & Caswell, 2004*).

Subtidal sessile communities are likely to be strongly affected by human activity in the marine environment and are economically and ecologically important. Nevertheless, some aspects of their dynamics remain poorly understood. Artificial structures such as offshore wind farms, oil rigs and docks (known collectively as ocean sprawl) can create new hard substrate, and thus act as stepping stones increasing connectivity between natural habitat patches (*Henry et al., 2018*). Subtidal sessile communities on structures such as offshore wind farms can affect other ecosystem components, with important socioeconomic consequences such as changes to fisheries yields (*Haraldsson et al., 2020*).

The development of these communities affects the design and operation of structures such as offshore oil rigs, but can also lead to commercially useful products such as shellfish and pharmaceuticals (*Page, Dugan & Piltz, 2010*). The temporal development and depth gradient patterns in temperate fouling communities are well known (*Whomersley & Picken, 2003*). Many aspects of such patterns can be understood in terms of the tradeoff between colonization rates and ability to compete for space (*Bracewell, Johnston & Clark, 2017*). However, there is evidence that factors other than space may sometimes be limiting in subtidal sessile communities, including food (*Svensson & Marshall, 2015*) and oxygen (*Ferguson, White & Marshall, 2013*), and in many cases we do not have a detailed understanding of the mechanisms controlling community dynamics. There are also methodological issues. Proportions of space occupied by sessile organisms are an example of compositional data. Naive analysis of relationships among the parts of a composition (such as between percentage cover of different groups of organisms) is misleading because of spurious correlation problems (*Aitchison, 1986*, pp. 48–50). This issue is sometimes overlooked, for example by ecologists attempting to infer competition from patterns in percentage cover (*e.g.*, *Willcox, Moltschaniwskyj & Crawford, 2008*). A key property of compositional data is that all relevant information is contained in logs of ratios of parts (*Aitchison, 1986*, chapter 4). Several important early examples of compositional data analysis are ecological (*e.g.*, *Billheimer, Guttorp & Fagan, 2001*; *Mosimann, 1962*) but compositional data analysis has been relatively little used by ecologists, other than those working on coral reefs (*e.g.*, *Gross & Edmunds, 2015*; *Vercelloni et al., 2020*) and microbiome data (*e.g.*, *Grantham et al., 2019*; *Silverman et al., 2019*).

Jellyfish polyps are an important but under-studied component of subtidal sessile communities. There is increasing evidence that jellyfish medusae play a key role in marine food webs (*Hays, Doyle & Houghton, 2018*). Demographic models suggest that the sessile polyp life stage of jellyfish can be very long-lived, and that polyp survival strongly affects population growth (*Goldstein & Steiner, 2019*). Ocean sprawl is thought to increase the availability of habitat for jellyfish polyps (*Duarte et al., 2013*). There is observational evidence for competitive and sometimes mutualistic interactions between jellyfish polyps and other sessile organisms, typically inferred from patterns in abundance on settlement panels or natural substrates (*e.g.*, *Colin & Kremer, 2002*, *Ishii & Katsukoshi, 2010*, *Rekstad et al., 2021*, *Watanabe & Ishii, 2001*, *Willcox, Moltschaniwskyj & Crawford, 2008*). However, experimental evidence is limited. For example, in an experimental manipulation of *Aurelia aurita* polyp density on settlement panels, high polyp densities were associated with reduced settlement of other organisms, and polyps were overgrown by other organisms (*Gröndahl, 1988*), although no data analysis was attempted. In addition, survival of *Cyanea nozaki* polyps was higher where the settlement of other organisms was reduced by mesh enclosures (*Feng et al., 2017*). Since most potential competitors are much larger than typical jellyfish polyps, it seems likely that if there is competition for space, it will be asymmetric, with jellyfish polyps affected by their potential competitors more strongly than *vice versa*. There is also evidence that polyps are more tolerant of hypoxia than many of their potential competitors, and this may affect the outcome of competition, with polyps doing better in low oxygen conditions near the bottom of the water column (*Ishii & Katsukoshi, 2010*). However, relatively little is known about the details of interactions between jellyfish polyps and other marine sessile organisms.

Here, we describe an experimental study of the interaction between *A. aurita* polyps and potential competitors on settlement panels in a brackish dock whose walls support a dense community of sessile organisms (*Chong & Spencer, 2018*; *Fielding, 1997*, chapter 4), dominated by green and red algae, solitary and colonial ascidians (*e.g.*, *Ascidiella aspersa*, *Botryllus schlosseri*, *Botrylloides* spp., *Ciona intestinalis*, *Clavelina lepadiformis*, *Molgula tubifera*, *Styela clava*), bryozoans (*Bugula* spp.), cnidarians (*Diadumene cincta*), mussels (*Mytilus edulis*) and sponges (*Halichondria* spp.). *Aurelia aurita* medusae are abundant in the summer, and polyps are found throughout the year, particularly towards the bottom of the dock walls. Oxygen concentrations are sometimes low at nearby sites, particularly close to the bottom in summer (*Fielding, 1997*, pp. 74–78). We determine the responses of the system to reduction in relative abundance of either *A. aurita* or potential competitors. We carry out these reductions at two depths, because it is plausible that differences in environmental conditions such as oxygen concentration affect the outcome of competitive interactions. We take two approaches to analysis of the data. First, we take a phenomenological approach, using a compositional manova model to analyze the effects of removal treatments and depth on relative abundances at the end of the experiment. We predict that removal of potential competitors will result in a relative increase in *A. aurita*, and that this increase will not depend on depth, because *A. aurita* polyps are relatively tolerant of low oxygen concentrations and often increase in abundance with depth. We also predict that removal of *A. aurita* may result in a relative increase in

potential competitors, but that this increase will be stronger at shallower depths, where oxygen is less likely to be limiting to potential competitors. However, it seems likely that competition between *A. aurita* and potential competitors is asymmetric, with potential competitors affecting *A. aurita* more than vice versa. Second, we take a more mechanistic approach, measuring interaction strengths between *A. aurita* and potential competitors using a series of models for community dynamics fitted to data. We determine whether the observed responses to manipulation can be generated by a model of preemptive competition for space, and whether this competition is asymmetric as predicted above.

## METHODS

### Experiment

#### Study site

The experiment was done in Salthouse Dock, Liverpool (53.4015° N, 2.9912° W), a semi-enclosed, brackish, non-tidal water body with stone walls and a depth of approximately 4 m, part of a dock system originally constructed in the 19th century, and redeveloped for recreational use in the 1980s (*Fielding, 1997*, pp. 11–14, 17). Permission to work at the site was given by the Canal and River Trust.

#### Settlement panels

Interactions between *A. aurita* polyps and other sessile organisms were investigated on 60 settlement panels (grey PVC, 100 mm × 100 mm × 5 mm, roughened to provide a better surface for colonization). Previous experiments showed that such PVC panels support a similar set of species to that found on the dock walls (*Maxatova, 2016*; *Presser, 2019*; *Sharpe, 2020*). Panels were suspended from a pontoon running along the dock wall in blocks of 6, with 3 in each block at 1 m and 3 at 3 m. Previous work has found substantial differences between dock wall communites at these depths (*Chong & Spencer, 2018*). The three panels at each depth were attached to the underside of a hardwood bar by a single stainless steel screw through the centre of each panel. A strip of lead along the underside of the bar ensured that panels always faced downwards. Bars were attached to the pontoon by 5 mm diameter nylon cords. Panels were suspended on 30 July 2019, a time of year when larvae of sessile organisms are usually abundant, and many *A. aurelia* medusae appeared ready to spawn. Human interference with panels was unlikely, because they were not readily visible from above and access to the pontoon was restricted to boat owners.

#### Treatments

PVC panels were assigned to one of three treatments: control (C), *A. aurita* polyp removal (A) and removal of potential competitors (O). Among the three panels in each block at each depth, one was assigned randomly to each treatment. In the A treatment, half of the *A. aurita* polyps on the underside of the panel were removed once a week by scraping with the tip of a plastic pipette. In the O treatment, every second individual or colony of each other species on the underside of the panel was removed using a paint scraper. Proportions removed were judged by eye. On one occasion (panel 2, 13 August 2019, the second week

of sampling), the A treatment was mistakenly applied to a control panel at 1 m depth. In the analyses described below, we treated this panel as a control when studying the final community, but included the A treatment in the second week of sampling when analysing temporal data.

### Sampling

Panels were sampled photographically every 7 days for 8 weeks (ending on 24 September 2019). Panels were pulled out of the water, placed face-up in a plastic box containing dock water, and photographed twice from a distance of approximately 100 mm using a Canon Powershot G10 14.7 megapixel digital camera (Canon Inc., Tokyo, Japan). Sampling using a stereo microscope would have improved the detectability of small organisms, but was not logistically feasible in the field. Panels other than those in the control group were photographed both before and after treatment, unless no relevant organisms were visible to remove (for example, no *A. aurita* polyps were visible in the first week of sampling). Dissolved oxygen, temperature and salinity were measured each week (except that no salinity measurements were taken in the fifth week) at both 1 and 3 m, using YSI 550 (oxygen) and 556 MPS (temperature and salinity) meters (YSI Inc., Yellow Springs, Ohio, USA). A Secchi disc was visible to at least 3.5 m in every week.

### Analysis of environmental data

Differences in dissolved oxygen, temperature and salinity between 3 and 1 m were investigated using central 95% credible intervals for the mean difference between depths in pairs of measurements from the same week. Under the assumption that differences between depths were independently and identically normally distributed, and with a noninformative uniform prior on the mean and log standard deviation, the standard one-sample *t*-interval is a central 95% credible interval for the mean difference between depths (*Gelman et al., 2003*, Section 3.2). The assumption of approximate normality was checked using *QQ*-plots, which did not reveal any major problem.

### Photograph analysis

Proportional cover of each taxon was estimated on each panel in each week by point counting. The sharpest photograph from each pair was selected, and the organism present (if any) at each of 100 randomly-located points recorded using JMicroVision version 1.3.1 (*Roduit, 2007*). The resolution of photographs was generally good enough to determine what organism was present, but when the organism present at a point could not be determined, the point was redrawn. The absence of macroscopic organisms was recorded as 'bare panel', which includes the presence of a biofilm of microorganisms. *A. aurita* polyps growing on potential competitors were recorded separately from those growing directly on the panel. Point count data were exported as ASCII text files and compiled into a single data set for statistical analysis. If a panel was not photographed before and after treatment (a control panel, or a treatment panel on which none of the target organisms were visible), the same point count data were used for before and after.
## Analysis of final composition

We used a Bayesian latent hierarchical compositional manova with a multinomial observation model to determine how final proportional cover was affected by treatments. A manova is the obvious way to examine patterns in multiple species, and a compositional approach is needed because we have relative abundance data, for which the standard vector addition and scalar multiplication operations used in manova are not appropriate. *Pawlowsky-Glahn, Egozcue & Tolosana-Delgado (2015)* is a good introduction to compositional data analysis. A multinomial observation model is the obvious choice for data derived from point counts. We analyzed the pre-treatment data from the final photographic sampling date, and included only *A. aurita* growing directly on panels, bare panel and other taxa contributing at least 20 points to the point count data for at least one panel: *Botrylloides* spp., *Bugula* spp. and *Molgula tubifera*. Together, these five taxa accounted for 90–100 points out of 100 on every panel in the pre-treatment point count data from the final week, and no other taxon contributed more than seven points on any panel. Compositional data analysis is subcompositionally coherent (*Egozcue & Pawlowsky-Glahn, 2011*, Section 2.3.2), which means that results for the subcomposition we studied do not depend on excluded taxa. We therefore analyzed final subcompositions of the form $c = (c_1, c_2, c_3, c_4, c_5)$, where parts one to five represent *A. aurita* on panel, bare panel, *Botrylloides* spp., *Bugula* spp. and *M. tubifera*, respectively. We represented these final subcompositions in isometric logratio (ilr) coordinates (*Egozcue et al., 2003*) using the contrast matrix described in the supporting information, Section S1.

Let $\mathbf{y}_{jkl}$ be the vector of point count data for the single panel from depth *j*, treatment *k*, block *l*, and let $n_{jkl}$ be the total number of points counted in this observation (between 90 and 100). We modelled these data using a Bayesian latent hierarchical compositional manova with a multivariate observation model:

$$
\begin{aligned}
\mathbf{y}_{jkl} &\sim \mathrm{multinomial}(n_{jkl}, \boldsymbol{\rho}_{jkl}), \\
\boldsymbol{\rho}_{jkl} &= \mathrm{ilr}^{-1}\Big(\boldsymbol{\mu} + \boldsymbol{\alpha}_j + \boldsymbol{\beta}_k + \boldsymbol{\gamma}_{jk} + \boldsymbol{\delta}_l + \boldsymbol{\varepsilon}_{jkl}\Big), \\
\boldsymbol{\delta}_l &\sim N(\mathbf{0}, \mathbf{Z}), \\
\boldsymbol{\varepsilon}_{jkl} &\sim N(\mathbf{0}, \Sigma).
\end{aligned}
\tag{1}
$$

Here, $\boldsymbol{\rho}_{jkl}$ is the vector of expected relative abundances for the panel from depth *j*, treatment *k*, block *l*. The isometric log transformation of $\boldsymbol{\rho}_{jkl}$ is a vector in $\mathbb{R}^4$, formed from the sum of an overall mean vector $\boldsymbol{\mu}$, the effect $\boldsymbol{\alpha}_j$ of depth *j*, the effect $\boldsymbol{\beta}_k$ of treatment *k*, the effect $\boldsymbol{\gamma}_{jk}$ of the interaction between depth *j* and treatment *k*, the effect $\boldsymbol{\delta}_l$ of block *l* and the effect $\boldsymbol{\varepsilon}_{jkl}$ of the panel from depth *j*, treatment *k*, block *l*. The block and panel effects are modelled hierarchically, drawn from 4-dimensional multivariate normal distributions with mean vector $\mathbf{0}$ and covariance matrices $\mathbf{Z}$ and $\Sigma$ respectively (independent of each other and of the explanatory variables). Note that $\boldsymbol{\rho}_{jkl}$ can be written in the simplex $\mathbb{S}^4$ as

$$
\boldsymbol{\rho}_{jkl} = \boldsymbol{\mu}' \oplus \boldsymbol{\alpha}_j' \oplus \boldsymbol{\beta}_k' \oplus \boldsymbol{\gamma}_{jk} \oplus \boldsymbol{\delta}_l' \oplus \boldsymbol{\varepsilon}_{jkl}',
\tag{2}
$$

where the primes indicate $\mathrm{ilr}^{-1}$ transformations of the corresponding parameters in $\mathbb{R}^4$, and $\oplus$ denotes the perturbation operator (*Aitchison, 1986*, p. 42). We coded treatment

effects as described in the supporting information, Section S2. Similar models have been used for effects of vegetation disturbance and predator manipulation on terrestrial arthropod communities (*Billheimer, Guttorp & Fagan, 2001*), effects of depth on community composition at our study site (*Chong & Spencer, 2018*), and effects of cyclones and bleaching on coral reef composition (*Vercelloni et al., 2020*).

We fitted the model using Bayesian estimation in cmdstan 2.23.0 (*Carpenter et al., 2017*), which implements a dynamic Hamiltonian Monte Carlo algorithm (*Hoffman & Gelman, 2014*). Details of priors are given in the supporting information, Section S3. Details of fitting, checking and calibration are given in the supporting information, Section S4.

We compared the ability to predict new observations between the full model and simpler models (without the interaction between depth and treatment, without depth, or without treatment) using leave-one-cluster-out cross-validation. The natural choice for "new observations" is a new block of panels, because a replication of the experiment would involve a new set of blocks, rather than new panels within existing blocks or new observations on existing panels. We therefore evaluated models based on marginal rather than conditional likelihoods with respect to block and panel effects (*Merkle, Furr & Rabe-Hesketh, 2019*). Details are in the supporting information, Section S5.

Our primary interest is in responses of *A. aurita*, bare panel and potential competitors as a whole, rather than variation within the subcomposition of potential competitors. Visualizing $\mathbb{S}^4$ is not easy, so we decomposed treatment effects into two orthogonal components, each of which can be represented in a ternary plot: effects on *A. aurita*, bare panel and potential competitors as a whole, and effects on the subcomposition of potential competitors (supporting information, Section S6).

We assessed the effects of potential competitors on *A. aurita* using differences in logit($A.aurita$) between potential competitor removal ($O$) and control ($C$) treatments. Similarly, we assessed the effects of *A. aurita* on potential competitors using differences in logit(potentialcompetitors) between *A. aurita* removal ($A$) and control ($C$) treatments, as described in the supporting information, Section S7.

## Models for community dynamics
### Basic model description

We will consider two state variables: the proportion of substrate $x$ filled by potential competitors such as ascidians and bryozoans (dimensionless) and the density $y_1$ of *A. aurita* polyps per unit area of substrate (numbers $L^{-2}$). Before collecting data we had planned to include a third state variable $y_2$ representing polyps on potential competitors. Some potential competitors provide suitable microhabitat for polyps (*e.g.*, *Rekstad et al., 2021*), and we have observed polyps on potential competitors in the past. However, in our data, there were very few polyps on potential competitors. We therefore do not consider $y_2$ in the main text, although we we describe the full model in the supporting information (Section S8). Our basic model allowed only preemptive competition for space between polyps and potential competitors. Preliminary analyses described below showed that this basic model could not reproduce the qualitative patterns found in experimental data, in

which polyps appeared to have positive effects on potential competitors. We therefore introduced a series of modifications after initial analysis of experimental data.

We treat both state variables and time $t$ (T) as continuous. For simplicity, we treat the dynamics of these variables (including the effects of removal treatments) as deterministic, and do not explicitly consider the spatial organisation of the system. A system of two ordinary differential equations is therefore a natural modelling approach. We treat the system as open, because we are modelling only the hard-substrate part of the ecosystem. We assume that polyps and potential competitors interact through preemptive competition for space. It is widely believed that space is often limiting for communities of sessile marine organisms on hard substrates (*Witman & Dayton, 2001*, p. 356). There is evidence that competition for food (*Svensson & Marshall, 2015*) and oxygen (*Ferguson, White & Marshall, 2013*) may also be important in fouling communities, but for simplicity we do not include these resources. The simplest plausible model is therefore

$$\frac{dx}{dt} = a_0(1 - x - \delta y_1) + a_1 x(1 - x - \delta y_1) + a_2 x, \tag{3}$$

$$\frac{dy_1}{dt} = b_0(1 - x - \delta y_1) + b_1 y_1(1 - x - \delta y_1) + b_2 y_1, \tag{4}$$

The processes included in this model are sketched in Fig. 1. This model is almost identical to a model for competition for space between branching and tabular corals (*Muko, Sakai & Iwasa, 2001*), except that we treat settlement rates as depending on the proportion of free space rather than the absolute amount of free space. We assume that larvae arrive at the same rate at all points in space, but only succeed in settling on free space, while *Muko, Sakai & Iwasa (2001)* presumably allow larvae to seek out only free space.

The dynamics of potential competitors are represented by Eq. (3). The positive parameter $a_0$ (T$^{-1}$) is the rate at which the proportion of unoccupied substrate is reduced by settlement of potential competitors, and the proportion of unoccupied substrate is $1 - x - \delta y_1$, where the positive parameter $\delta$ is the area of substrate occupied per polyp (numbers$^{-1}$L$^2$). The positive parameter $a_1$ (T$^{-1}$) is the proportional rate at which the proportion of unoccupied substrate is reduced by growth of potential competitors already on the substrate. The negative parameter $a_2$ (T$^{-1}$) is the proportional rate at which the proportion of unoccupied substrate is increased by death of potential competitors already on the substrate. The dynamics of polyps (Eq. (4)) have the same form as Eq. (3). The parameters are the proportional rate of settlement of polyps on unoccupied substrate ($b_0$, positive, numbers L$^{-2}$T$^{-1}$), the proportional rate of increase of polyp number on substrate by budding of polyps on substrate ($b_1$, positive, T$^{-1}$) and the proportional death rate of polyps on substrate ($b_2$, negative, T$^{-1}$).

We measure interaction strengths using the community matrix of partial derivatives of proportional rates of change with respect to relative abundances of polyps and potential competitors. This is an appropriate choice of interaction strength measurement for our experiment, because it does not require the assumption of equilibrium (*Laska & Wootton,*

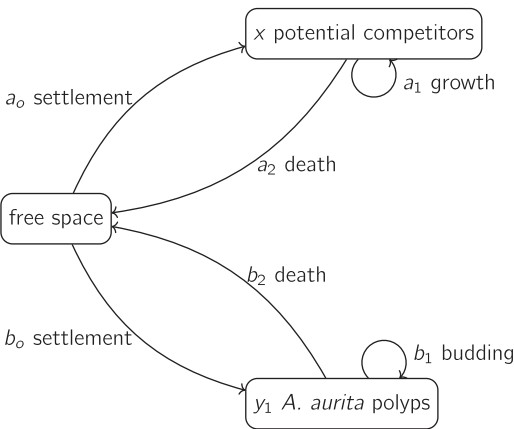

**Figure 1 A basic model for the dynamics of polyps and potential competitors, as in Eqs. (3) and (4).** Full-size ⬛ DOI: 10.4282/sosj.20.7

*1998*). We include effects on settlement, because we want to measure the overall effects on proportional rates of change of relative abundances. However, if we wanted a measure of habitat quality alone, it would be more appropriate to exclude effects on settlement (*Drake & Richards, 2018*). In the supporting information (Section S9), we show that the signs of the elements in the community matrix are

$$\begin{pmatrix} - & - \\ - & - \end{pmatrix},\tag{5}$$

where element $(1, 1)$ is the intra-group effect of potential competitors, element $(1, 2)$ is the proportional effect of polyps on potential competitors, element $(2, 1)$ is the proportional effect of potential competitors on polyps, and element $(2, 2)$ is the intra-group effect of polyps. Thus, each group of organisms in the model has overall negative intra-group density dependence, and potential competitors and polyps on substrate have negative effects on each other.

### Mechanisms for positive effects of polyps on potential competitors

Inspection of experimental data suggested positive effects of polyps on potential competitors. The basic model only allows negative effects (Expression 5, element $(1, 2)$). We therefore considered four mechanisms by which positive effects could occur: facilitation of settlement, facilitation of growth, overgrowth of polyps by potential competitors, and protection from predators. Each requires a change to Eq. (3) and one new parameter, and overgrowth also requires a change to Eq. (4). For each, we briefly outline possible biological justifications. In the supporting information, Section S11, we show that each can give a positive effect of polyps on potential competitors, for some values of $x, y_1$ and parameters.

We modelled facilitation of settlement as follows:

$$\frac{dx}{dt} = (a_0 + m_0 \delta y_1)(1 - x - \delta y_1) + a_1 x(1 - x - \delta y_1) + a_2 x,\tag{6}$$

where the positive parameter $m_0$ ($\text{T}^{-1}$) represents the increase in settlement rate of potential competitors for a unit increase in the proportion of space occupied by polyps. Settlement by one species may facilitate settlement by other species through changes to the properties of the substrate, including hydrodynamics and the microbial biofilm (*Wieczorek & Todd, 1998*). A linear effect is the simplest plausible model.

Similarly, we modelled facilitation of growth as follows:

$$\frac{dx}{dt} = a_0(1 - x - \delta y_1) + (a_1 + m_1 \delta y_1)x(1 - x - \delta y_1) + a_2 x, \tag{7}$$

where the positive parameter $m_1$ ($\text{T}^{-1}$) represents the increase in rate of growth of potential competitors onto unoccupied space for a unit increase in the proportion of space occupied by polyps. Mechanisms for facilitation of growth are less obvious than those for facilitation, but it is known that *A. aurita* polyps support a microbial community distinct from that of their surroundings (*Weiland-Bräuer et al., 2015*), and that ascidians can retain particles as small as bacteria (*Petersen, 2007*), although the extent to which the *A. aurita* polyp microbiome can affect the microbiome ingested by filter-feeders is unknown. Again, a linear effect is the simplest plausible model.

Overgrowth of polyps by potential competitors requires modelling the loss of polyps due to overgrowth, as well as the gain in space occupied by potential competitors:

$$\frac{dx}{dt} = a_0(1 - x - \delta y_1) + a_1 x(1 - x - \delta y_1) + a_{1,y_1} x y_1 + a_2 x, \tag{8}$$

$$\frac{dy_1}{dt} = b_0(1 - x - \delta y_1) + b_1 y_1(1 - x - \delta y_1) - \frac{a_{1,y_1}}{\delta} x y_1 + b_2 y_1, \tag{9}$$

where the positive parameter $a_{1,y_1}$ ($\text{numbers}^{-1}\text{L}^2\text{T}^{-1}$) represents the rate at which potential competitors overgrow polyps. Temporal and spatial variation in polyp abundance suggest that *A. aurita* competes with other sessile organisms (*Watanabe & Ishii, 2001*; *Ishii & Katsukoshi, 2010*). It seems plausible that potential competitors, particularly the larger ones, could overgrow *A. aurita* polyps. As above, a linear effect is the simplest plausible model.

Protection from predators requires a slightly different approach, because the final term in Eq. (3), representing death of potential competitors, must always be negative. We used the modification.

$$\frac{dx}{dt} = a_0(1 - x - \delta y_1) + a_1 x(1 - x - \delta y_1) + a_2 e^{-m_2 \delta y_1} x, \tag{10}$$

where the positive parameter $m_2$ (dimensionless) represents the rate at which increases in the proportion of space covered by polyps reduce the death rate of potential competitors. Predation can have substantial effects on the abundance of early life stages of solitary and colonial ascidians (*Osman & Whitlatch, 2004*). In contrast, relatively few species appear to feed on *A. aurita* polyps, and some of those that do show evidence of being deterred by nematocysts in polyp tentacles (*Takao, Okawachi & Uye, 2014*). Thus, it is plausible that *A. aurita* tentacles could deter predators from feeding on other species. A brief justification

for the modelling approach is as follows. Assume that the proportion of space swept by polyp tentacles or within which a predator is close enough to polyps to be deterred visually is proportional to the proportion of substrate occupied by polyps ($\delta y_1$), with constant of proportionality $k$ (dimensionless). Call this the proportion of space affected by polyps. This involves the implicit assumption that no part of the substrate is affected by more than one polyp, which will be approximately true when polyps occupy only a small proportion of space. Suppose that a predator moves at a constant speed across the surface in a randomly-oriented straight line in order to consume a potential competitor. Then the expected proportion of its path affected by polyps is $k\delta y_1$ (*Kaiser, 1983*). Suppose that a predator will feed only if it does not have a physical or visual encounter with a polyp (a deterrence event), and that these events happen at rate 0 in areas unaffected by polyps, and rate $p$ (dimensions $T^{-1}$) in areas affected by polyps. Then the overall rate will be $(1 - k\delta y_1) \cdot 0 + k\delta y_1 p = k\delta y_1 p$. Let a unit of time be the time needed for the predator to travel the full path needed to feed. Then the probability that no deterrence events happen during this time is $e^{-kp\delta y_1}$. Let death happen at rate $a_2$ when $y_1 = 0$. Then the death rate in the presence of predators will be $a_2 e^{-kp\delta y_1}$, which is the exponential model above, with $m_2 = kp$. Note that this does not explicitly account for other causes of death. However, unless $m_2$ is large, the death rate will not be close to zero when $\delta y_1 = 1$.

### Application to experimental data

We fitted versions of Eqs. (3) and (4), with each of the modifications in Section "Mechanisms for positive effects of polyps on potential competitors" in turn, to the experimental data from all weeks and panels, as described in the supporting information, Sections S12, S13 and S14.

### Visualization of results

For each model, we plotted posterior mean predicted relative abundances against time in a typical panel from each combination of treatment and depth, with 95% highest posterior density credible bands. A typical panel is one having the most common series of treatment applications for the combination of treatment and depth: no treatment applications in the control; treatment applications from the third week onwards in the *A. aurita* removal treatment; treatment application from the second week onwards in the potential competitor removal treatment.

To understand the effect of *A. aurita* polyps on the proportional rate of change of potential competitors, we plotted the posterior mean of this effect on a grid of points in the simplex, for each model at each depth, and overlaid trajectories of posterior mean predicted relative abundances for typical panels from each combination of treatment and depth.

Comparison of fitted models suggested that estimates of the proportion $r_A$ of *A. aurita* removed in the $A$ treatment differed between models. As a visual check on the plausibility of each model, we plotted post-treatment against pre-treatment sample proportions of space covered by *A. aurita* each week in the *A. aurita* removal treatment, along with lines

through the origin with slope $1 - r_A$ (with 95% highest posterior density credible bands), representing predictions from each model.

As noted above, experimental data suggested positive effects of polyps on potential competitors. In order to rule out the possibility that these effects arose from accidental removal of potential competitors in the *A. aurita* removal treatment, we plotted post-treatment against pre-treatment sample proportions of space covered by potential competitors each week in the *A. aurita* removal treatment. If *A. aurita* removal is not also removing potential competitors, we would expect points in these plots to fall along a line through the origin with slope 1.

## RESULTS

### Environmental data

There was little evidence for systematic differences in dissolved oxygen (supporting information, Fig. S5A, mean difference $-0.73$ mg L$^{-1}$, central 95% credible interval $[-1.74, 0.29]$ mg L$^{-1}$) or salinity (supporting information, Fig. S5C, mean difference 0.09 psu, central 95% credible interval $[-0.06, 0.23]$ psu) between 3 m and 1 m. However, water at 3 m was systematically colder than water at 1 m (supporting information, Fig. S5B, mean difference $-0.26$ °C, central 95% credible interval $[-0.47, -0.05]$ °C).

### Panel communities

All panels were initially empty. Early colonizers included colonial arborescent bryozoans (*Bugula* spp.), colonial ascidians (*Botrylloides* spp. and *Botryllus schlosseri*) and *Aurelia aurita* polyps, all of which appeared within the first 2 weeks. The solitary ascidian *Molgula tubifera* had become abundant within 4 weeks of the start of the experiment. The solitary ascidian *Ascidiella aspersa* began to appear after 7 weeks. By the final week of the experiment, the organisms occupying at least one randomly-chosen sampling point out of 100 on at least one panel were (in descending order of proportion of space occupied) *Molgula tubifera*, *Bugula* spp., *Botrylloides* spp., *Aurelia aurita* and *Ascidiella aspersa*. Examples of panels from all treatments from the final week of the experiment are shown in Fig. 2. Many of the *Molgula tubifera* had died and dropped off the panels by 29 October 2019, roughly 1 month after the end of the experiment, so the final week of the experiment may be close to the peak of competition for space.

### Analysis of final composition

All the results for final composition reported below are based on a model with depth and treatment effects, but without an interaction between them. The difference in expected log predictive density for a new block between the full model and a model with no interaction was negligible (Table 1, row 2), and the graphical and numerical summaries discussed below were similar between models with and without an interaction. In contrast, models without an interaction and a removal treatment effect, or without an interaction and a depth effect, were much worse than the model with depth and removal treatment effects but no interaction (Table 1, rows 3 and 4). Parameter estimates for the selected model are given in the supporting information, Table S1.

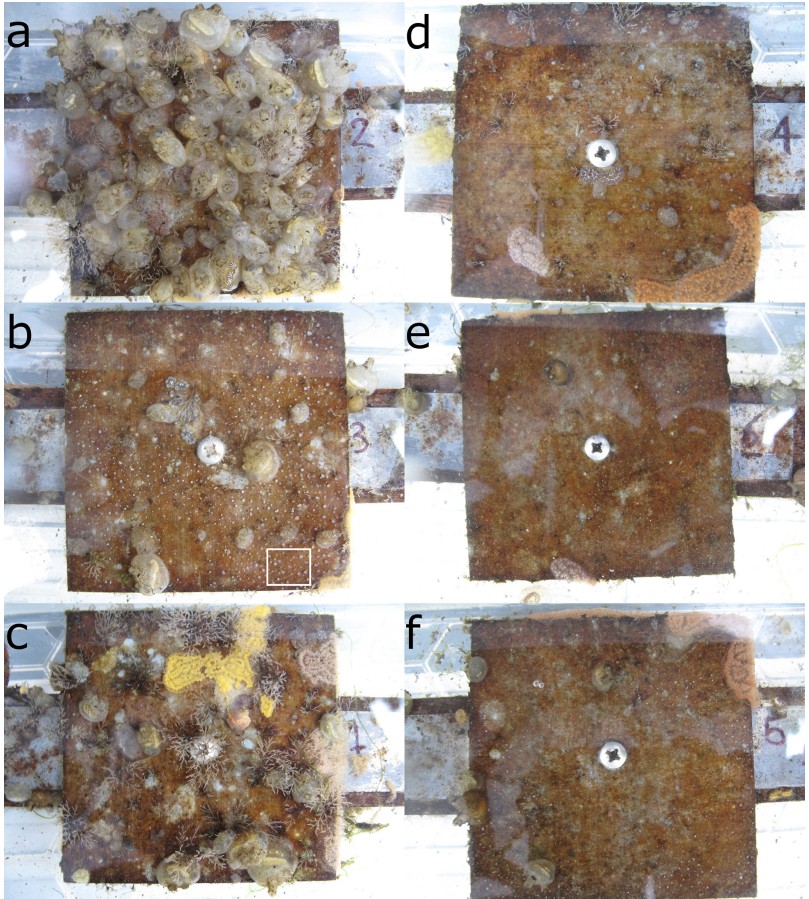

**Figure 2 Panel photographs from the end of the experiment (2019-09-24, pre-treatment) at 1 (A–C) and 3 m (D–F).** Photos A and D are controls (*C*), B and E are potential competitor removal treatment (*O*), and C and F are *A. aurita* removal (*A*). The panels shown here are a single block. The white rectangle in the bottom right of B encloses an area dominated by *A. aurita* polyps. A closeup of the bottom right corner of B, appparently showing overgrowth of polyps by *Botrylloides sp.*, is shown in the supporting information, Fig. S9. Note that the *A* treatment was mistakenly applied to the control panel in a on 2019-08-13.                                Full-size ⊡ DOI: 10.4282/sosj.20.7

   Overall, panels at 3 m had relatively more *A. aurita* and bare panel, and less space occupied by potential competitors, than panels at 1 m (Fig. 3A, filled *vs* open large circles, Figs. 2D–2F *vs* 2A–2C). At each depth, there was relatively little difference between the control and *A. aurita* removal treatments (Fig. 3A, green *vs* orange large circles are close together, with overlapping 95% credible regions, Figs. 2A *vs* 2C and 2D *vs* 2F), although there was a tendency towards relatively more bare panel in the *A. aurita* removal treatment. Composition in the potential competitor removal treatment appeared distinct from the other two treatments, with relatively less space occupied by potential competitors and slightly more *A. aurita* (Fig. 3A, purple *vs* green and orange large circles, Figs. 2B and 2E). Treatment and depth had little effect on the subcomposition of potential competitors (Fig. 3B), with overlapping 95% credible regions for all combinations, although there was some tendency for panels at 3 m to have relatively more *Botrylloides* spp. and less *Bugula* spp., compared to those at 1 m (Fig. 3B, filled *vs* open circles).

**Table 1 Model selection for compositional manovas, data from final week, based on expected log predictive density for a new block.** Each row shows the difference in expected log predictive density ($\Delta\text{elpd}_{\text{loco}}$) between a given model and the best model in the top row, and the standard error (SE) of the difference. Formulae in the Model column give the effect of a combination of depth $j$ and removal treatment $k$ in the simplex ($\phi'_{jk}$) in terms of depth effect $\alpha'_j$, removal treatment effect $\beta'_k$ and interaction $\gamma'_{jk}$. Expected log predictive density was estimated for a new block of panels by leave-one-cluster-out cross-validation, with Monte Carlo integration over the distributions of block and panel effects.

| Model | $\Delta\text{elpd}_{\text{loco}}$ | SE |
|---|---|---|
| No interaction: $\phi'_{jk} = \alpha'_j \oplus \beta'_k$ | 0 | 0 |
| Full: $\phi'_{jk} = \alpha'_j \oplus \beta'_k \oplus \gamma'_{jk}$ | −25.0 | 20.2 |
| No interaction, no removal treatment effect: $\phi'_{jk} = \alpha'_j$ | −1005.4 | 66.9 |
| No interaction, no depth effect: $\phi'_{jk} = \beta'_j$ | −1510.9 | 102.1 |

*Aurelia aurita* responded positively to removal of potential competitors at both 1 m (Fig. 4A, purple: posterior mean logit difference 1.68, 95% credible interval $(1.15, 2.21)$) and 3 m (Fig. 4B, purple: posterior mean logit difference 0.50, 95% credible interval $(0.07, 0.93)$), although the posterior mean effect was further from zero at 1 m than at 3 m. Unexpectedly, potential competitors responded negatively to removal of *A. aurita* at both 1 m (Fig. 4A, orange: posterior mean logit difference −0.66, 95% credible interval $(−1.12, −0.20)$) and 3 m (Fig. 4B, orange: posterior mean logit difference −0.64, 95% credible interval $(−1.10, −0.18)$).

Both among-panel variation and among-block variation (described by the covariance matrices $\Sigma$ and $\mathbf{Z}$ respectively) were non-negligible. In particular, there was variation at panel level in the geometric mean of potential competitors relative to *A. aurita* and bare panel (supporting information, Fig. S6: green ellipses are stretched out towards the gm(potentialcompetitors) vertex). Within the subcomposition of potential competitors, panel-level variation appeared to be more important than block-level variation (supporting information, Fig. S7: green ellipses generally lie outside orange ellipses).

## Models for community dynamics

Polyps of *A. aurita* first appeared 2 weeks after panels were put in the water, but their relative abundance remained low throughout the experiment (Fig. 5A, faint lines). Throughout, they tended to have higher relative abundance at 3 m than at 1 m (Fig. 5A: faint solid lines generally above faint dashed lines). By the end of the experiment, they tended to have the highest relative abundance in the potential competitor removal treatment and the lowest relative abundance in the *A. aurita* removal treatment (Fig. 5A: faint purple lines generally above faint green lines, and faint green lines generally above faint orange lines, by the end of the experiment). The relative abundance of bare panel was clearly higher at 3 m than at 1 m by the end of the experiment (Fig. 5B: faint solid lines above faint dashed lines). Conversely, the relative abundance of potential competitors was clearly higher at 1 m than at 3 m by the end of the experiment (Fig. 5C: faint dashed lines generally above faint solid lines). As noted above in the analysis of final composition, there was an unexpected tendency for the relative abundance of potential competitors to be

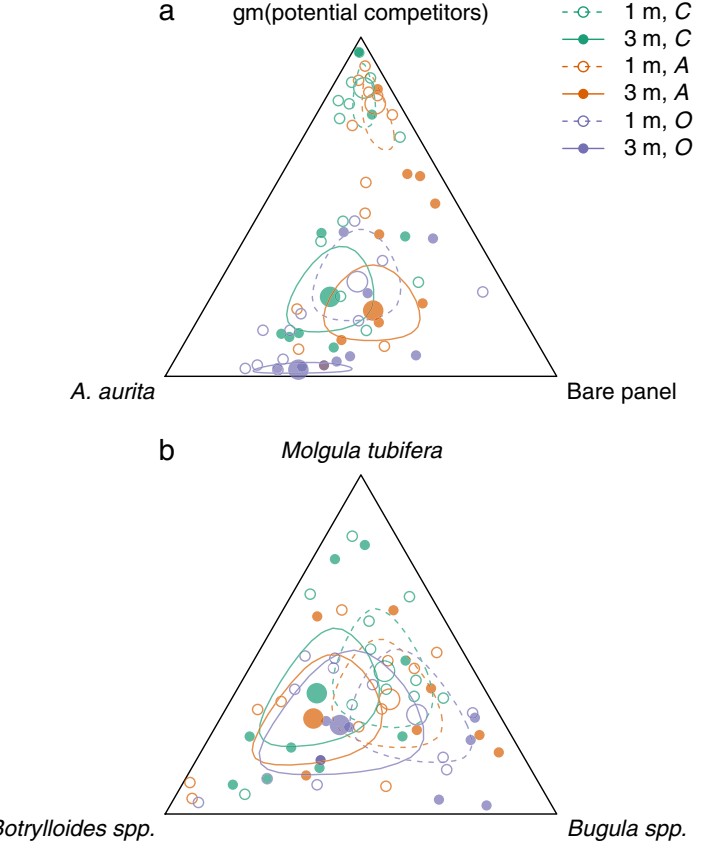

**Figure 3 Effects of removal treatments and depth on community composition at the end of the experiment.** a: orthogonal projection onto the 2-simplex with parts representing *A. aurita*, bare panel and gm (potential competitors), where gm () denotes the geometric mean. b: orthogonal projection onto the subcomposition of potential competitors. Open circles and dashed lines are from 1 m, filled circles and solid lines from 3 m. Colours represent removal treatments: control (*C*) green, *A. aurita* removal (*A*) orange, potential competitor removal (*O*) purple. Small circles represent observations (final week, pre-treatment), large circles estimated treatment effects from manova. Lines are the boundaries of 95% highest posterior density credible regions. For plotting, zero counts are replaced by 1/2.
Full-size ⬛ DOI: 10.4282/sosj.20.7

higher in the controls than the *A. aurita* removal treatment by the end of the experiment (Fig. 5C: faint green lines tend to be above faint orange lines; Fig. 4: orange density curves).

The overgrowth model partially reproduced the unexpected pattern of potential competitors having higher relative abundance in the controls than the *A. aurita* removal treatment, but only at 3 m (Fig. 5C: solid green line above orange green line). Furthermore, the estimated effect of *A. aurita* on the proportional growth rate of potential competitors was positive for the overgrowth model at 3 m (supporting information, Fig. S8B), but negative at 1 m (supporting information, Fig. S8A), for all compositions. Although we did not attempt any systematic direct observations of overgrowth, it does appear that at least *Botrylloides* is able to overgrow *A. aurita* polyps (supporting information, Fig. S9). There was some evidence from cross-validation that the overgrowth model was better than all the others, although the difference in expected log predictive density from the next best model was less than two standard errors of the difference (Table 2). At 1 m, where the proportion

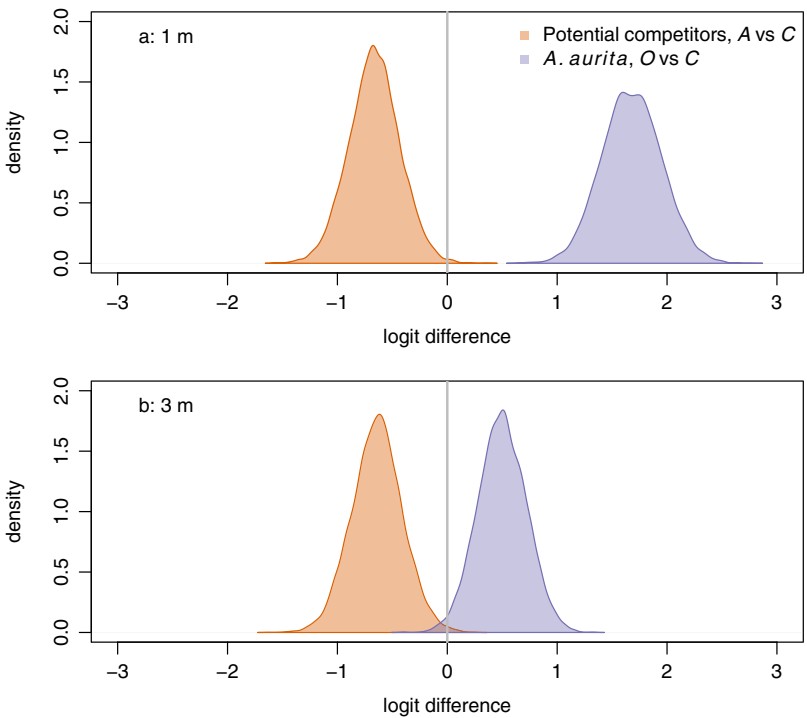

**Figure 4 Responses of potential competitors to removal of _A. aurita_ (orange), and of _A. aurita_ to removal of potential competitors (purple) at 1 m (A) and 3 m (B), estimated from manova on final week, pre-treatment data.** The response of potential competitors is the difference in logit potential competitors between the _A. aurita_ removal (A) and control (C) treatments. The response of _A. aurita_ is the difference in logit _A. aurita_, between the potential competitor removal (O) and control (C) treatments. Posterior distributions of responses represented using kernel density estimates. Vertical grey lines indicate null response.  Full-size 🖼 DOI: 10.4282/sosj.20.7

of space covered by polyps was low, the estimated rate of overgrowth of polyps by potential competitors in the overgrowth model was small compared to the rate of growth of potential competitors over bare panel (supporting information, Table S2, $a_{1,y_1^*}$ and $a_1$ respectively). However, at 3 m, the estimated rate of overgrowth of polyps by potential competitors was much larger than the estimated rate of growth of potential competitors over bare panel. Models other than overgrowth were more or less indistinguishable from each other in terms of expected log predictive density for a new observation (Table 2), and none of them reproduced the unexpected pattern of higher relative abundance of potential competitors in the controls than the _A. aurita_ removal treatment (supporting information, Figs. S10–S13). The only other model to produce a positive effect of _A. aurita_ on the proportional growth rate of potential competitors was the settlement facilitation model, but only in a very small set of compositions with low relative abundance of potential competitors, high relative abundance of bare panel, and moderately low relative abundance of _A. aurita_ (supporting information, Fig. S8G, very small blue area in bottom right corner). This positive effect in the settlement facilitation model has little relevance to predicted dynamics, because typical trajectories (supporting information, Fig. S8G, lines) do not pass through it. All models reproduced the other qualitative features of the observed time series described above.

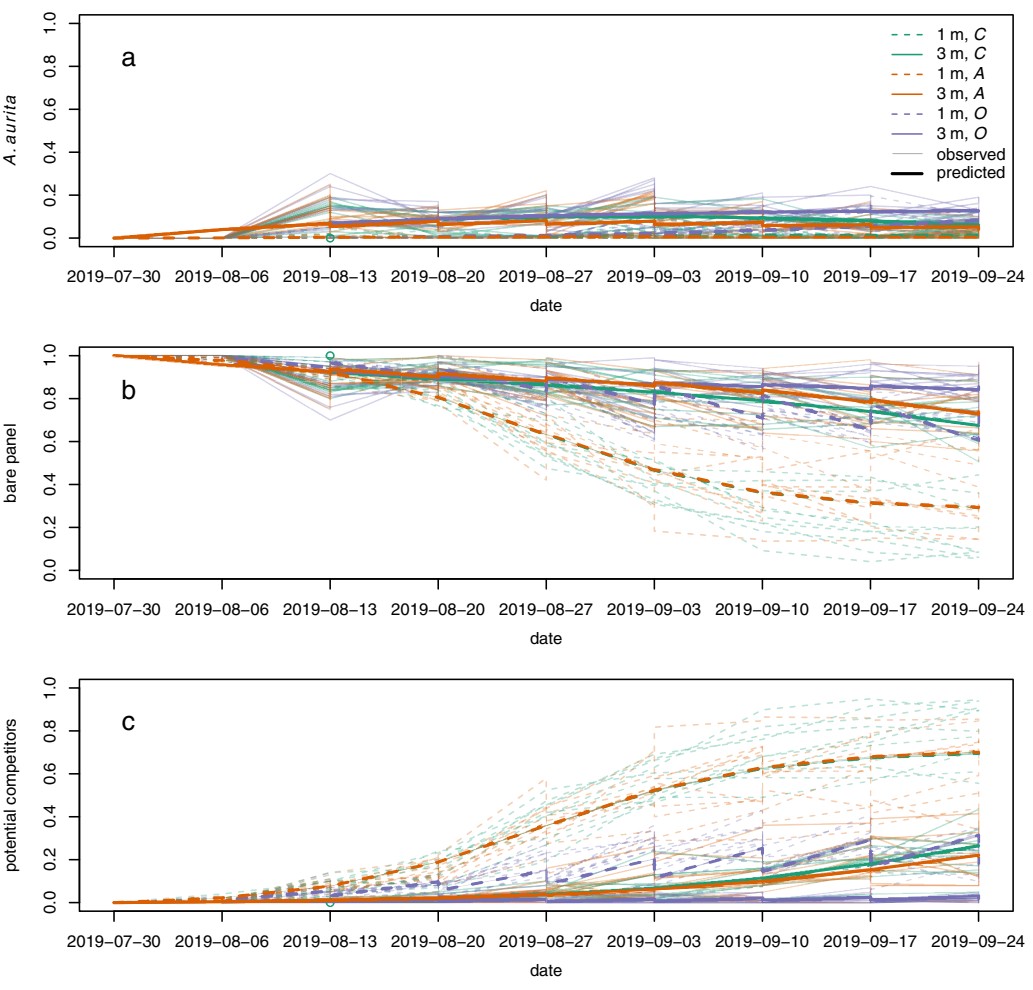

**Figure 5 Modelled (bold lines, overgrowth model) and observed (faint lines) time series for proportional cover of (A) *A. aurita*, (B) bare panel and (C) potential competitors.** Each bold line is the posterior mean for a typical panel from a combination of treatment and depth. Each faint line is the time series of observations from a single panel. Dashed lines represent panels at 1 m, and solid lines panels at 3 m. Colours represent treatments: control (*C*) green, *A. aurita* removal (*A*) orange, potential competitor removal (*O*) purple. 95% highest posterior density credible bands are shown for modelled time series, but are usually too narrow to be visible. Panels were put in the water on 2019-07-30. Open green circle on 2019-08-13: control panel at 1 m to which *A* treatment was mistakenly applied on the second sampling date. Full-size ◩ DOI: 10.4282/sosj.20.7

The estimated proportions removed in treatments in the overgrowth model were approximately 0.2 for *A. aurita* in the *A* treatment and 0.42 for potential competitors in the *O* treatment (Table S2, $r_A$ and $r_O$ respectively). These were clearly below the target values of 0.5 for each, but well above zero. Estimates for other models were very similar for $r_O$, but larger for $r_A$. Plots of post- against pre-treatment proportions of space filled by *A. aurita* in the *A* treatment did not strongly distinguish between the plausibility of estimates of $r_A$ from different models, although if anything models other than overgrowth appeared to represent the post- *vs* pre-treatment *A. aurita* data better, and there was a tendency for all models to underestimate the proportion of *A. aurita* removed for larger pre-treatment proportions of space occupied by *A. aurita* (supporting information, Fig. S14: points for

**Table 2 Model selection for ordinary differential equation models based on expected log predictive density for a new observation calculated using Pareto-smoothed importance sampling.** Each row shows the difference in expected log predictive density ($\Delta$elpd$_{loo}$) between a given model and the best model in the top row, and the standard error (SE) of the difference.

| Model | $\Delta$elpd$_{loo}$ | SE |
|---|---|---|
| Overgrowth | 0 | 0 |
| Protection | −32.0 | 18.1 |
| Basic | −32.0 | 18.2 |
| Settlement facilitation | −33.4 | 17.4 |
| Growth facilitation | −34.3 | 16.1 |

larger pre-treatment values generally lay below lines through the origin with slope $1 - r_A$). There was no evidence that potential competitors were being accidentally removed along with *A. aurita*: post- and pre-treatment proportions of space filled by potential competitors in the *A* treatment lay approximately on a line through the origin with slope 1 (supporting information, Fig. S15).

The overgrowth model appeared moderately plausible, but there was still room for improvement. Posterior predictive simulation from the overgrowth model (supporting information, Fig. S16) showed that although this model captured some of the main features of dynamics as noted above, it underestimated the amount of variability among panels within a treatment combination, compared to the real data (Fig. 5, wide spread of faint lines for each combination of line style and colour). In particular, this model did not reproduce the large variation in the proportion of space filled by potential competitors on the real panels at 1 m in the *A* and *C* treatments, at the end of experiment (Fig. 5C, faint lines, *vs* supporting information, Fig. S16C, orange and green dashed lines). This failure is perhaps not surprising, because our dynamic models were deterministic, while variation among panels may be strongly driven by stochastic variation in settlement. On simulated data, although there was no evidence of gross errors, 95% HPD intervals did not often contain the true parameter value for the parameters $a_0$ at 1 m (supporting information, Fig. S17A, 3/10 simulated data sets), $a_1$ at 1 m (supporting information, Fig. S17C, 0/10 simulated data sets), $a_2$ at 1 m (supporting information, Fig. S17E, 0/10 simulated data sets), $\delta b_0$ at 3 m (supporting information, Fig. S17H, 3/10 simulated data sets) and $b_2$ at 3 m (supporting information, Fig. S17L, 3/10 simulated data sets). In all but the first of these cases, the posterior modes tended to be pulled towards zero compared to the true true parameter values, which may indicate a strong influence of the half-normal priors with modes at zero. Furthermore, the posterior distributions for the proportional death rates of potential competitors $a_2$ at 3 m (supporting information, Fig. S17F) and of polyps $b_2$ at 1 m closely matched the prior distributions, suggesting that there was little information in the data on these parameters. This may be a consequence of the low proportional cover of potential competitors at 3 m and of polyps at 1 m, respectively (Fig. 5: C, faint solid lines, and A, faint dashed lines, respectively). Thus, even this most successful model should be viewed as at best a rough approximation to the processes generating the data.

## DISCUSSION

As predicted, removal of potential competitors resulted in a relative increase in *A. aurita*, which did not appear to depend on depth. This is consistent with previous observational (*e. g.*, *Watanabe & Ishii, 2001*; *Colin & Kremer, 2002*; *Willcox, Moltschaniwskyj & Crawford, 2008*; *Ishii & Katsukoshi, 2010*) and experimental (*Gröndahl, 1988*; *Feng et al., 2017*) studies. Below, we suggest that this interaction may, over time, moderate the response of jellyfish populations to the creation of new habitat such as offshore wind farms. Unexpectedly, removal of *A. aurita* resulted in a relative decrease in potential competitors, which did not appear to depend on depth. Although we predicted an asymmetric interaction, we did not predict a reversal of sign. The lack of dependence on depth may be because oxygen was not limiting in our study system during the experiment, although it might be at other times. Our models of competition for space were only partially able to generate the observed pattern. The most successful of these models suggested overgrowth of *A. aurita* by potential competitors as a possible mechanism, but only generated the observed pattern at 3 m, and gave only a modest improvement in ability to predict new observations. Below, we suggest some possible approaches to understanding this unexpected result. Finally, *Roughgarden (1986)* suggested that subtidal communities similar to our study system may be lattice communities, in which density-independent mortality is low relative to the rate of settlement, and in which growth stops and density-dependent mortality is low once space is exhausted. In a separate classification, *Roughgarden (1986)* also suggested that such subtidal communities are CNP communities (Closed because most of the organisms involved have relatively short dispersal distances, and limited by space, which is Not Partitionable). We evaluate the evidence for these suggestions, and the implications for future approaches to community dynamics in subtidal hard substrate communities.

Removal of potential competitors resulted in a relative increase in *A. aurita*. Both physical pre-emption of space ("founder control", as in our basic model) and overgrowth ("dominance", as in our overgrowth model) might contribute to this effect (*Yodzis, 1986*). *A. aurita* is a rapid colonizer of empty space. Thus, we expect that when new habitat is created by coastal or offshore development, there will be a rapid initial increase in polyp density, ephyra production and medusa abundance. Our experimental evidence for a negative effect of potential competitors on relative abundance of *A. aurita* polyps implies that as potential competitors increase in relative abundance over a time scale of years to decades (*e.g.*, *Whomersley & Picken, 2003*), relative abundance of *A. aurita* polyps will decrease again, so that the increase in medusa abundance may be transient (*Feng et al., 2017*). However, sessile organisms including solitary ascidians and *M. edulis* provide suitable substrate for *A. aurita* polyps (*Rekstad et al., 2021*). There were few *A. aurita* polyps on these organisms in our experiment, but this is not the case in every year (M. Spencer, personal observation). Extensive settlement of polyps on potential competitors could change the sign of effect of potential competitors (Section S8), and thus alter the long-term consequences of habitat creation for jellyfish populations.

Removal of *A. aurita* polyps resulted in an unexpected relative decrease in potential competitors, at both depths. The evidence from this experiment was clear, but it will be important to determine whether it replicates across years and study locations. In particular, the substantial mortality of the potential competitor *M. tubifera* observed after the end of the experiment was unexpected, as the closely-related *M. manhattensis* is thought to live for about 1 year (*Zvyagintsev, Sanamyan & Koryakova, 2003*). Thus, replication will be important to establish whether the outcome was a consequence of unusual conditions towards the end of the experiment. Although we do not have an explanation for the effect of *A. aurita* on potential competitors, there are some possibilities that seem unlikely. We do not think this is likely to be an experimental artefact, because panels were removed from the water in sets of three (one from each treatment, arranged in a random order) and placed together in a tank of dock water for photography. Other than the treatments, all panels experienced the same conditions. Accidental removal of potential competitors along with *A. aurita* polyps also seems unlikely. Polyps were removed individually by hand, and the appearance of polyps is quite different from that of potential competitors. Furthermore, comparison of proportions of space filled by potential competitors before and after polyp removal suggests that accidental removal was negligible (Fig. S15). Any mechanism that depends on depth seems unlikely, because in the analysis of final composition, a model without an interaction between treatment and depth had similar ability to predict new observations to a model with such an interaction. We did not observe low-oxygen events during the experiment, although it is possible that some such events might have occurred between sampling dates. Settlement facilitation can be important in fouling communities (*e.g.*, *Dean & Hurd, 1980*), but our dynamic models did not support this explanation, and the experiments in *Dean & Hurd (1980)* did not rule out other mechanisms. Nevertheless, it is possible that removal of biofilm along with *A. aurita* polyps could have influenced settlement of potential competitors. Although some of our potential competitors are known to be vulnerable to predators, particularly when small (*e.g.*, *Botrylloides*, *Vieira, Flores & Dias, 2018*), and the stinging tentacles of polyps might deter predators, a dynamic model with protection from predators did not perform better than the basic model. Growth facilitation might plausibly occur through the distinct microbiome of *A. aurita* polyps (*Weiland-Bräuer et al., 2015*), but again this was not supported by the dynamic models. The dynamic models suggested that enhanced overgrowth of *A. aurita* polyps by potential competitors compared to growth onto bare panel was the most plausible mechanism. However, the details of how this mechanism might operate remain unclear, and even our overgrowth model did not capture the positive effect of *A. aurita* polyps on potential competitors at 1 m. The sea anemone *Metridium senile* can have short-term positive effects on other sessile organisms, perhaps through disrupting boundary layer flow (*Nelson & Craig, 2011*). It is possible that a dense carpet of *A. aurita* polyps could have a similar effect, leading to increased food supply to nearby potential competitors and subsequent overgrowth. However, the low relative abundance of *A. aurita* makes this an unlikely explanation in the 1 m treatment. The *A. aurita* polyp microbiome (*Weiland-Bräuer et al., 2015*) might plausibly affect overgrowth rather than growth onto bare panel. However, it is important not to overinterpret the evidence for

mechanisms from our dynamic models, given the modest differences in expected log predictive density between the overgrowth model and other models. Further experiments might therefore be the best way to distinguish between possible mechanisms. For example, detailed observation of community development on panels in the laboratory could confirm that the apparent effect is real, whether it is caused by overgrowth, and would allow manipulation of factors such as larval supply and predation. If settlement facilitation is important, the positive effect of polyps on potential competitors would disappear if there was no settlement, while if protection from predators is important, the positive effect would disappear when predators were excluded. An artefact of biofilm removal along with polyp removal could be ruled out using a removal-control treatment in which the polyp removal method was applied to areas of bare panel. Distinguishing between overgrowth and growth facilitation would require measurement of the rates at which potential competitors grow onto bare panel and over polyps. More generally, it seems somewhat unrealistic that in our most successful model, the effect of *A. aurita* on the proportional population growth rate of potential competitors did not depend on the relative abundance of *A. aurita* (Section S11.3). Although this property is shared by the Lotka-Volterra model (and is therefore less surprising than it initially appears), it would be worth designing experiments with a sufficiently wide range of *A. aurita* relative abundances that more flexible models could be evaluated.

Two classifications of competitive communities may help in understanding the nature of interactions in this system. *Roughgarden (1986*, pp. 509–513) suggested that subtidal communities might often be lattice communities, with low density-dependent and density-independent mortality rates, high settlement rate relative to density-independent mortality rate, growth that stops when space is exhausted, and close to 100% cover. Our results do not support this suggestion. For both *A. aurita* polyps and potential competitors, estimated density-independent mortality in the best-fitting dynamic model had a substantially greater magnitude than settlement (supporting information, Table S2, settlement rates $a_0$, $\delta b_0$, density-independent mortality rates $a_2$, $b_2$, in potential competitors and *A. aurita* polyps respectively), although these estimates should be interpreted cautiously, given the extent to which they depend on the choice of suitable model structure, including simplications such as using deterministic models for underlying dynamics. The best-fitting model had overgrowth of *A. aurita* polyps by potential competitors, so that growth does not necessarily stop when space is exhausted. Except in the controls at 1 m, most panels had a large proportion of free space at the end of the experiment, and our communities appear to be a closer match to the high free-space community type, with low settlement rate relative to density-independent mortality and limitation by recruitment (*Roughgarden, 1986*,p. 512). Surveys of nearby dock walls suggest that a substantial proportion of free space will remain in the long term (*Chong & Spencer, 2018*). *Roughgarden (1986*, p. 515) also classified competitive communities by whether the system is open or closed, and whether the limiting resource is partitionable. It was suggested that subtidal communities might be CNP systems (Closed, due to short dispersal distances, but with space being Not Partitionable). However, it does not make sense to model experimental systems of settlement panels, or newly-constructed structures

such as offshore wind farms, as closed systems. Thus, ONP (Open, but with a Non-Partitionable limiting resource) seems a more appropriate classification for such communities. Despite their limited success in reproducing the patterns seen in our experiments, models with the structure that we used, and those of *Muko, Sakai & Iwasa (2001)*, are a natural choice for ONP systems. If they are of the high free-space type, for which stochastic fluctuations in settlement rate can strongly affect relative abundances, it is likely that a stochastic differential equation formulation, with temporal variation in settlement rates, would be a productive approach. Nevertheless, it is reasonable to hope that deterministic models such as those considered here will be of some use in understanding the qualitative behaviour of ONP systems.

In conclusion, although potential competitors for space such as ascidians and bryozoans had the expected negative effect on *A. aurelia* polyps, the positive effect of *A. aurita* polyps on potential competitors was unexpected and remains unexplained. A combination of new experiments (involving detailed monitoring of growth rates onto bare panel and polyps, and manipulation of larval supply and predation) and mathematical models is needed to confirm that this is a real effect, and to determine the mechanism behind it. These results are important because they suggest that interspecific interactions in a canonical example of a competitive system are more complex than is generally believed.

## ACKNOWLEDGEMENTS

We are grateful to Ariel Greiner, Xikun Song and an anonymous reviewer for comments on the manuscript, Hannah Bills for help with field work, and to Les Connor, Carmel Pinnington and Phil Robson for technical support.

### Funding

This work was funded by the University of Liverpool's Herdman Endowment. The funders had no role in study design, data collection and analysis, decision to publish, or preparation of the manuscript.

### Grant Disclosures

The following grant information was disclosed by the authors:
University of Liverpool's Herdman Endowment.

### Competing Interests

The authors declare that they have no competing interests.

### Author Contributions

- Jade Boughton conceived and designed the experiments, performed the experiments, authored or reviewed drafts of the article, and approved the final draft.
- Andrew G. Hirst conceived and designed the experiments, authored or reviewed drafts of the article, and approved the final draft.

- Cathy H. Lucas conceived and designed the experiments, authored or reviewed drafts of the article, and approved the final draft.
- Matthew Spencer conceived and designed the experiments, performed the experiments, analyzed the data, prepared figures and/or tables, authored or reviewed drafts of the article, and approved the final draft.

## Field Study Permissions

The following information was supplied relating to field study approvals (*i.e.*, approving body and any reference numbers):

Permission for field experiments was given by the Canal and River Trust.

## Data Availability

The raw data and code are available in the Supplemental File.

## Supplemental Information

Supplemental information for this article can be found online at http://dx.doi.org/10.7717/peerj.14846#supplemental-information.

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
