# Peer review of "Negative and positive interspecific interactions involving jellyfish polyps in marine sessile communities"

_PeerJ, doi:10.7717/peerj.14846_

## Round 0.1 · original submission · Minor Revisions

I believe this study is innovative with solid methodology and detailed data. Both reviewers evaluated the study positively, but they also had some targeted changes. Please follow the reviewers' comments and revise each of them.

·

Basic reporting

The manuscript is novel in studying the jellyfish polyps and their potential competitors by adopting integrating experiments and modeling. I am sorry that the modeling is out of my research scope, and I have recommended two experts in numeric ecology modeling. Generally, I support the publication of this research after revisions. The Introduction and Discussion could be shortened and highlight the main findings. Here I am listing some points when I read the manuscript from the biological aspects.

Method
Lines 115-121 Please move these important research backgrounds to the Introduction.
Line 121: “the bottom of the dock walls”. Please specify the component, cement?
Line125: PVC, is this comparative to the natural filed in Salthouse Dock or previous field observations using (plastic) panels (Lines 115-121)? Please explain.
Lines 127, 155: 1m and 3m, are there significant differences for the fouling biota? Three meters seems to be too shallow.
Line 135: PVC panels were…
Line 147: I have used the Canon 100 mm lens, I doubt some mm-sized scyphozoan polyps or pedal cysts may be overlooked. Such kind of polyps should be observed under a stereo microscope.
Line 163: If the image is based on Canon 100 mm lens, is the resolution clear enough for counting polyps?

Results
Line 377: Bugula spp. or Bugula sp.? If the manuscript are investigating the same species of the genus Bugula, please use Bugula sp.; spp. and sp. should not be shown in italic, please revise throughout.
Line 409: A. aurita should be Aurilia aurita.

Discussion
Lines 533, 534: This may also be attributable to the removal of biofilms when A. aurita polyps were removed. Biofilms play important roles in inducing larval settlement of invertebrate larvae.
Line 598, 600: I am not sure whether it is convenient to list the large ascidians and bryozoans as the competitors of the tiny scyphozoan polyps. Ascidians and bryozoans are filter-feeding organisms. However, scyphozoans are carnivorous. Some hydrozoan polyps (hydroids) may be real competitors, however, the present study may have overlooked them, or they do not exist in the present simulation biota.

Sincerely,
Xikun Song
Xiamen University

Experimental design

See Part I.

Validity of the findings

See Part I.

Reviewer 2 ·

Basic reporting

Summary of the article:
The authors have applied robust and standard statistical models to the compositional data of a sessile colony on experiment panels subjected to different taxa-removal treatments. Both phenomenological approach (statistical inference and comparison of logits) and a more mechanistic approach based on ODE modelling has suggested that polyps of Aurelia aurita may have positive effect on its potential competitors for space and this effect might be attributable to the overgrowth by its competitors, as suggested by the lower elpd value of the overgrowth model. Otherwise, the experiment results conform to the expectation that removal of potential competitors always has positive effects on A.aurita polyps.
I agree that the aim of the study is well justified, and the interesting results do suggest that some the interspecific interactions in highly competitive ONP systems may be more complicated than previously conceived.
I do not have strong opinions regarding the overall validity of the experiment design (which seems fine and standard methods were used) or the conclusion; the following general comments may better improve the quality of the manuscript and I recommend its publication once the following issues are addressed.

General comments:

I. Typos, etc.:
line 432:
a wrong sentence: the relative abundance of potential competitors was clearly higher at 1 m (!), not at 3 m as written in the manuscript.
line 573:
high “settlement (rate)" ? a word ("rate") seems to be missing.

Experimental design

II. Method and data analysis (2 issues):
Issue 1: how species were removed
Take the A-treatment at 3 m as an example. The ratio between the post-treatment competitor coverage and the pre-treatment value (Table 2 in the attached PDF) seems quite scattered, In many cases, the ratio is much greater than 1 or less than 0.5. This might suggest that the counting was not very accurate; how would this affect subsequent analyses? The ratio of some blocks is more scattered than others, can this kind of potential “counting/treatment error” be taken into account by the block-effect in manova?

Issue 2: parameter estimation
Overall, I am happy with data analyses and how they are presented; here are sufficient and detailed materials both in support of the robustness of the methods, and materials revealing the potential problems/limitations of the current model, e.g. Figure S14, S15, S17, presented in a lucid manner (Figure S16 seems blurred; not sure if it is a problem with my own screen). However, the estimate of rA still surprises me:
line 465, main text:
"These were clearly below the target of 0.5 for each, but well above zero"; the 0.2 seems too low and the 95% credible highest density region does not cover 0.5, either. I have compiled the raw data and you can find them in Table 2 of the attached PDF file. Here is large scatter in the Post/Pre ratio of polyps in A treatment at 3 m; however, when averaging across all the data, indeed we see rA ~ 0.50. I wonder why the model has missed it so much.
I fed the raw data and the overgrowth ODE model into SCEM-UA algorithm (Vrugt, 2003: A Shuffled Complex Evolution Metropolis algorithm for optimization and uncertainty assessment of hydrologic model parameters), and noticed that estimated rA is in the range of 0.43~0.55, a1_y1, the overgrowth factor, takes values similar to the reported ones; though other parameters can take values very different from those reported in the manuscript.
Equifinality could be an answer to this issue; in view of this, maybe you could remind readers that interpretations of the magnitude, sign etc. of the parameters (e.g. line 572~578) need to be taken cautiously/with a pinch of salt. Or you might consider showing robustness of results by applying a different, supplemental parameter estimation method to your data (then see how much the estimated parameter values differ from those in this version of manuscript), if time permits.

Validity of the findings

III. Discussion and interpretation of results (three issues):

Issue 1: in ODE models, the positive effects of polyps (y) on potential competitors seem to be independent of polyps density or only occurs when y is low, though cited evidence that might support the overgrowth/settlement facillitation mechanisms may suggest that a reasonable density of polyps is needed
line 557: The abundance of Au at 1m was quite low (always lower than 4%). I find it interesting if it should have such obvious positive effects on the growth of competitors; in Nelson and Craig (2011) cited in the manuscript, here was considerable anemone density (20~25%) on the panel they deployed for their short-term field experiment;
line 356 in Supplemental Material: the positive effect only depends on some parameter values and the magnitude of x.
However, "dense carpet of ..." (line 560) implies that a certain abundance of au is needed for initiating the cited mechanisms and indeed it may require a reasonable abundance of polyps for them to sufficiently disturb the flow or change the microbiome; so I wonder how the c(1,2) element in line 356 does not depend on y. Similarly, for other models, y1_star < (1-x)/2 will give positive c(1,2) (e.g. line 338 in Supplemental Material); though the existence of an upper limit of y1_star that gives positive c(1,2) is quite plausible, it seems a bit suprising that a lower limit is absent. Possible mechanisms?
I’m not sure if there are other examples where a very low abundance of species A (e.g. a few isolated patches) can positively affect its competitor, B; and it could be more convincing if you cite these examples (if any), or explain more how the positive effect may be independent of the abundance of polyps (i.e. the c(1,2) element is independent of y).

Issue 2: Duration of the experiment; abrupt change in taxa's densities near the end of the experiment
I see abrupt, sudden drops in competitors' density in the last week (A-treatment, 3 meter), disrupting the previous trends (see Figure 1 in the attached PDF file). Though this is not so obvious for the A-treatment at 1 meter.

In some cases, the competitors in the final composition would have take up a higher proportion than in control group, if there had not been these sudden drops.
Were these catastrophic crashes, with a deterministic mechanism, or were they largely due to stochasticity? if the experiment had been extended for one more week, would they grow back to a higher
density? This feature in the data might undermine the overall interpretation of the experiment results: if it were indeed driven by some randomness and could be quickly replenihed by recruitment
over the ensuing week (thus, there being little difference between C and A treatment), then much of the discussion regarding the positive effects of polyps on competitors may need to be reconsidered;
I would appreciate it if possible implications of these sudden drops may need to be better clarified; yet I do understand that, given the existing data, it is hard to distinguish between true "crash" from randomness.

Issue 3: line 583 ~ 585; line 595 ~ 597
“Although a short term experiment …. are appropriate” (line 583 ~ 585) seems to suggest that the dynamic models in the manuscript are properly chosen, so their prediction of high percentage of free space is reliable despite the short duration of the experiment. However, I wonder if this is the actual meaning of the authors, whether the correct selection of model really needs to be implied here.
This statement in line 583 ~ 585 becomes more confusing when one comes across line 595 ~ 597; the observational evidence in line 586 and the argument that the system is more similar to ONP systems seem to suggest that line 595 ~ 597 is what authors really want to say (the system is ONP, and large free space suggests that stochasticity should be taken into account, and the adopted models may not be that appropriate); if this is the case, then temporal variation in settlement rate will be the more reasonable model specification, contradicting the statement in line 583 ~ 585.
I think line 583 ~ 585 can be removed; the cited Chong and Spencer (2018) is sufficient and good enough to show that large free space will persist in the long term. Then you will not rely on the parameter estimation from short-term models (conditioned on the very appropriate selection of model structure, which is an “assumption” there) to infer the long term composition. At the end of this paragraph, line 595~597 can stay, suggesting that a stochastic model may be more productive; though I think in this case, you might also want a concluding sentence there in defence of the “good but not perfect” deterministic model.

Annotated reviews are not available for download in order to protect the identity of reviewers who chose to remain anonymous.

---

## Round 0.2 · Minor Revisions

As you can see, the reviewers and I think you have done a good job of responding to most of the comments. However, reviewer 2 still has his comments on a small number of issues. I hope you will be able to address them properly or respond to them soon.

·

Basic reporting

Sincerely thanks for the substantial revisions by authors. All of My concerns have been well resolved. I hope every success of all authors.

Experimental design

Accepted as is.

Validity of the findings

Accepted as is.

Reviewer 2 ·

Basic reporting

I'm happy with the changes made and the responses given by the authors.

Experimental design

I think figure S14 actually highlights the large deviation of scatter points from the predicted 1-rA line... this is what prompted me to delve deeper into your dataset; but I do think the new added statement regarding the deterministic nature in modelling the possibly highly stochastic removal operation. The derivation of the SCEM-UA algorithm did not require that the errors be Gaussian, a different appropriate likelihood function may just be used (I used the likelihood function for the multinomial distribution; yet indeed, I have not treated heteroscedasticity properly in my likelihood function. The outgrowth model is the only case where SCEM-UA algorithm gave a rA estimates very different from those reported in the Supplemental Material. I have found this surprising. But we may leave this issue as it is.

As for equifinality, the concept was proposed in 80s in the field of hydrology, and discussed in details (with a focus on its interpretation and treatment in practice) in, e.g. "Equifinality, data assimilation, and uncertainty estimation in mechanistic modelling of complex environmental systems using the GLUE methodology" (Beven and Freer, 2000); it is totally different from the aliasing problem in sampling. It is not just about the end-state: "In environmental modeling studies, and especially in hydrological modeling, two models are equifinal if they lead to an equally acceptable or behavioral representation of the observed natural processes". In the field of hydrology, people are faced with calibrating highly nonlinear models against partly-observed trajectories of river runoff/tracer concentration, which is a task similar to the one of determining the parameters of each population model and comparing these models. I still maintain that it is a phenomenon almost universal to any open-system model described by a system of ODE/PDE and involve a relatively large number of parameters. This is the reason why I do not find interpreting parameter values of these complicated models (the overgrowth model has 8 parameters) very convincing. If you agree, I would appreciate it if a caveat regarding interpreting parameter values and equifinality is added. Otherwise, I regard this (and accept it as a reasonable one) as a minor disagreement in our opinions on how a model should be used in exploring new mechanisms and proposing new theories.
The model maybe good at replicating the trajectory of the observation data points (which is the case for many operational models, they are useful and we use them to make forecasts), yet it requires more theoretical analyses and experimental work to really validate the possibly new mechanisms revealed in them...

Validity of the findings

I am happy with the response, no more problems.

Additional comments

no more problems.

---

## Round 0.3 · accepted · Accept

I confirm that the authors have responded well to all of the reviewers' comments and that the quality of the manuscript has been greatly improved.
Thank all the authors for their efforts.